Analysis of the characteristic patterns and risk factors impacting the severity of intraoperative hypothermia in neonates

Dai Kun 1
Liu Yuanling 2
Qin Lijiao 1
Mai Jiaxuan 3
Xiao Jingjing 1
Ruan Jing 1 gdsfyrj@163.com
1 Department of Nursing, Guangdong Women and Children’s Hospital , Guangzhou, Guangdong , China
2 Administration Office, Guangdong Women and Children’s Hospital , Guangzhou, Guangdong , China
3 Neonatal Surgery Department, Guangdong Women and Children’s Hospital , Guangzhou, Guangdong , China
Fujioka Kazumichi
Electronic publication date: 2024 Dec 16
Publication date: 2024
Volume: 12
Electronic Location ID: e18702
Received 2024 Jun 7; Accepted 2024 Nov 22
Copyright: © 2024 Dai et al.
Copyright year: 2024
Copyright holder: Dai et al.
License: This is an open access article distributed under the terms of the Creative Commons Attribution License, which permits unrestricted use, distribution, reproduction and adaptation in any medium and for any purpose provided that it is properly attributed. For attribution, the original author(s), title, publication source (PeerJ) and either DOI or URL of the article must be cited.
License URL: https://creativecommons.org/licenses/by/4.0/

Keywords: Neonates, Surgery, Hypothermia, Risk factors, Characteristic patterns

Funding: The authors received no funding for this work.

==============================
Background

Although maintaining a stable body temperature during the perioperative period is crucial for the recovery of neonates, hypothermia frequently occurs during surgical procedures in this vulnerable population. A comprehensive analysis of intraoperative details, including medical history and monitoring, is therefore essential for understanding temperature variations and identifying risk factors for severe hypothermia.

Objective

In this study, we delineated the characteristic patterns of intraoperative temperature fluctuations in neonates and determined the risk factors impacting the severity of hypothermia.

Methods

We conducted a retrospective, single-center study, enrolling 648 subjects who underwent surgery under general anesthesia and collected demographic, perioperative, and intraoperative data.

Results

Intraoperative hypothermia occurred in 79.17% of the neonates. Significant risk factors for severe hypothermia included surgery type (OR, 1.183; 95%, CI [1.028–1.358]; p = 0.018), preoperative weight (OR, 0.556; 95% CI [0.412–0.748]; p < 0.01), infusion and transfusion volume (mL/kg) (OR, 1.011; 95% CI [1.001–1.022]; p = 0.018), and duration of hypothermia (OR, 1.011; 95% CI [1.007–1.014]; p < 0.01). Preterm neonates experienced a greater temperature drop than did full-term neonates. The nadir of intraoperative temperature occurred approximately 90 min after surgery, followed by a brief stabilization period and a slow recovery process.

Conclusion

The significant incidence of intraoperative hypothermia in neonates highlights the need for efficient strategies that reduce both the frequency and severity of this condition.

Introduction

Perioperative hypothermia is defined as a drop in core temperature below 36 °C (96.8 °F) during surgery (Sessler, 1997). Neonates are easily affected by external environmental temperatures due to their underdeveloped thermoregulatory center, relatively large body surface area, and lack of brown adipose tissue—all of which can lead to hypothermia (Ruan et al., 2024). Hypothermia in neonates exerts a significant impact on their bodies, increasing the risk of pressure ulcers (Ahtiala, Laitio & Soppi, 2018), postoperative cardiovascular events (Sessler, 2021), coagulatory abnormalities (Trckova & Stourac, 2018), surgical site infections (Guest, Vanni & Silbert, 2004), wound disruption (Görges et al., 2019), pain (Mendonça et al., 2021), increased bleeding (Rajagopalan et al., 2008), adverse effects on anastomotic healing (Ju et al., 2021), neonatal necrotizing enterocolitis, acute renal failure, intracranial hemorrhage, and even death (Ruan et al., 2024). Research indicates that the incidence of hypothermia in newborns during surgery can be as high as 82.83% (Zhao et al., 2023), with consequences that include prolonged hospital stays and increased healthcare costs (Takauji et al., 2021; Zhao et al., 2023), generating great challenges to the early postoperative recovery of neonates.

The regulation of body temperature is a complex process that involves temperature receptors, central mechanisms of temperature regulation, and effector responses (Scott et al., 2015), and general anesthesia can seriously impair the body’s thermoregulatory function while lowering the threshold for vasoconstriction and tremors. After anesthesia induction, the threshold for vasoconstriction and tremors falls, leading to a gradual weakening of the body’s self-regulatory capability. Due to this reduced regulatory ability, body heat is initially redistributed from the core to the periphery; and then heat-loss exceeds metabolic heat production, resulting in hypothermia (Sessler, 2016). In full-term newborns—and even in slightly premature infants—central thermoregulatory controls such as sweating and vasoconstriction thresholds are apparently intact (Sessler, 2008). However, these intact thresholds do not counteract the physiological differences—including reduced subcutaneous fat and a greater surface area-to-volume ratio—thereby increasing the risk of hypothermia. During general anesthesia, adults experience a characteristic pattern of hypothermia, with an initial rapid drop in core temperature followed by a slower decline and eventual stabilization (Sessler, 2016). However, studies on the temperature pattern of neonates during surgery remain limited.

Hypothermia contributes to various surgical complications, including surgical site infections, increased need for oxygen, altered pharmacokinetics of medications, impaired coagulation, and cardiac arrhythmias (Beedle et al., 2017)—negatively affecting early postoperative recovery of neonates. Researchers are currently primarily identifying risk factors for intraoperative hypothermia in neonates (Morehouse et al., 2014), with only a few studies addressing the severity and characteristic patterns of this condition. In this study, we analyzed the patterns and risk factors associated with intraoperative hypothermia in neonates and provided critical insights into the surgical experiences and insulation strategies in operating rooms, particularly in developing countries with limited resources. Our goal is to improve perioperative care for neonates by addressing this research gap, potentially reducing the frequency and severity of intraoperative hypothermia and its related complications.

Materials and Methods

Study design

The study was registered on the Chinese Clinical Trial Registry website (https://www.chictr.org.cn/bin/project/edit?pid=203701) with registration number ChiCTR2300074666 and adhered to the STROBE guidelines (Vandenbroucke et al., 2014). We conducted this investigation at a pediatric hospital in South China, with the operating room on the second floor and the Neonatal Intensive Care Unit (NICU) on the sixth. Newborns were transported from the NICU to the operating room in bassinets, with consistent warming measures in place to minimize temperature variations.

For this retrospective analysis, we adopted a convenience sampling method, sourcing data from our electronic medical record system. Eligible patients were identified using their unique patient ID numbers. Inclusion criteria were neonates aged 0–28 days who underwent surgery from 1 January 2021 to 28 February 2024. The exclusion criteria were as follows: (1) a history of hyperthyroidism, hypothyroidism, or other endocrine disorder affecting body temperature; (2) abnormalities in temperature regulation such as malignant hyperthermia; (3) surgeries involving low-temperature treatment, such as those involving extracorporeal circulation; (4) use of preoperative medicine that affects core temperature, including NSAIDs; and (5) a lack of core-temperature monitoring or loss of temperature data. We performed a multivariate analysis, aiming for a sample size of at least 500, with approximately 48 observations per estimated parameter to ensure adequate reliability for the variables under study.

Data collection

The data for this investigation comprised clinical characteristics that were collected and chosen in collaboration with nursing professionals and anesthesiologists. Data collection was conducted by investigators or proficient research assistants. The data-gathering processes were evaluated in a preliminary study of 100 patient charts at the hospital before the primary collection. Prior to the training of data collectors, minor adjustments that we deemed necessary were made to the data-gathering instrument. The software Epidata 3.1 was adopted for the purposes of gathering and organizing data. We obtained the data for these factors from the nursing, surgical, and anesthetic records for each individual surgical procedure.

Demographic and surgical characteristics

We initially collected information regarding the demographics of patients and their preoperative and intraoperative characteristics. The gathered data were classified into the following categories: (1) patient demographic characteristics such as sex, age, and preoperative weight; (2) preoperative parameters such as vital signs and infection indicators; and (3) intraoperative parameters that included the patient’s American Society of Anesthesiologists (ASA) grading, surgery type, emergency or planned surgery, surgical method, duration of anesthesia, volume of infusions and blood transfusions, intraoperative blood loss and other outputs, diagnosis, temperature records, and duration of hypothermia.

Upon arrival at the operating room, standard monitoring procedures were performed; these included monitoring electrocardiogram (ECG), measuring arterial blood pressure (ABP) and noninvasive blood pressure (NIBP), and determining peripheral blood oxygen saturation (SpO2). The anesthesia protocol involved midazolam (0.1 mg/kg), fentanyl citrate (3–5 μg/kg), and rocuronium (0.6–1.0 mg/kg) for induction; followed by sevoflurane (1–2%) for maintenance (Zhang et al., 2023). Intraoperative hypothermia was defined as a core body temperature below 36 °C, with classifications as mild (35–35.9 °C), moderate (34–34.9 °C), and severe (<34 °C) (Campbell et al., 2015). Throughout the operation, the patient’s core temperatures were continuously measured using either an esophageal or nasopharyngeal temperature probe. The flushing solution was heated to 37 °C as standard nursing practice, and selective active warming was applied when equipment permitted. Hypothermia was diagnosed if the core temperature fell below 36 °C according to our electronic monitoring system. The temperature in the operating room was maintained in a range of 25–27 °C (WHO, 1997; Black & Maxwell, 2024).

Statistical analysis

We adopted R 4.2.0 software to conduct statistical analysis and generate visual representations. Categorical variables are presented numerically and as proportions, and continuous variables are indicated by their mean (M) and standard deviation (SD). Independent-sample t tests were executed to compare the two groups. Values that did not follow a normal distribution were depicted as the median and interquartile range (M [Q]), and counting data are expressed as n (%) with the χ2 test employed to compare groups. We exploited multivariate logistic regression to examine the risk factors and consequences linked to hypothermia in surgical procedures. Statistical significance was determined by a p-value of < 0.05.

Ethical considerations

Approval for this investigation was granted by the Ethics Committee at Guangdong Women and Children’s Hospital (Ethical Application Ref: 202301230). Because the data for this study were obtained from electronic medical record systems, informed consent was not required.

Results

Comparison between hypothermic and non-hypothermic groups

Figure 1 shows the procedure for enrolling patients, and Table 1 displays a comparison of the baseline clinical data between the non-hypothermic group and the hypothermic group. We enrolled a total of 648 neonates: 386 boys (59.57%) and 262 girls (40.43%); the mean age of the neonates was 7.09 ± 7.31 days, the average gestational week was 36.88 ± 3.03 weeks, and the average weight was 2.76 ± 0.73 kg. The surgery type was general surgery in 507 cases (78.24%), urologic surgery in 31 cases (4.78%), cardiothoracic surgery in 74 cases (11.42%), and other surgeries in 36 cases (5.56%); 513 cases (79.17%) experienced intraoperative hypothermia. We noted significant differences between the two groups in gestational week, age, preoperative weight, preoperative temperature, red blood cell (RBC) count, hemoglobin (HB), type of surgery, duration of anesthesia, infusion and transfusion volume (mL/kg), blood loss (mL/kg), and hospital stay (p < 0.05).

Figure 1 Flowchart summarising the steps used for patient selection.

Table 1 Comparison of clinical baseline data in the non-hypothermic and hypothermic groups.

Variables	Total	No-hypothermia group	Hypothermia group	p	
n (%)	648	135 (20.83)	513 (79.17)		
Patient characteristics					
Sex, n (%)				0.83	
Male	386 (59.57)	82 (60.74)	304 (59.26)		
Female	262 (40.43)	53 (39.26)	209 (40.74)		
Gestational week, mean (SD)	36.88 (3.03)	37.44 (2.77)	36.73 (3.08)	0.01	
Age, mean (SD)	7.09 (7.31)	8.67 (8.37)	6.68 (6.96)	<0.01	
Preoperative weight, mean (SD)	2.76 (0.73)	2.95 (0.77)	2.72 (0.71)	<0.01	
Preoperative evaluation parameters				
Initial vital signs				
Preoperative temperature, mean (SD)	36.32 (0.33)	36.45 (0.24)	36.27 (0.34)	<0.01	
Laboratory test results					
WBC, mean (SD)	13.43 (5.89)	13.17 (5.61)	13.49 (5.96)	0.59	
RBC, mean (SD)	4.40 (0.81)	4.23 (0.79)	4.44 (0.82)	0.01	
HB, mean (SD)	152.29 (28.95)	146.89 (30.69)	153.71 (28.33)	0.01	
Intraoperative evaluation parameters				
Surgery type, n (%)				0.01	
General surgery	507 (78.24)	97 (71.85)	410 (79.92)		
Urology Surgery	31 (4.78)	4 (2.96)	27 (5.26)		
Cardiothoracic surgery	74 (11.42)	20 (14.81)	54 (10.53)		
Others	36 (5.56)	14 (10.37)	22 (4.29)		
Surgical method, n (%)				0.27	
Open surgery (Thoracotomy & laparotomy)	267 (41.20)	50 (37.04)	217 (42.30)		
Endoscopic surgery (Thoracoscope & laparoscope)	253 (39.04)	52 (38.52)	201 (39.18)		
Others	128 (19.75)	33 (24.44)	95 (18.52)		
ASA, n (%)				0.67	
I/II	186 (28.70)	41 (30.37)	145 (28.27)		
III	413 (63.73)	82 (60.74)	331 (64.52)		
IV	49 (7.56)	12 (8.89)	37 (7.21)		
Planned surgery, n (%)				1	
No	393 (60.65)	82 (60.74)	311 (60.62)		
Yes	255 (39.35)	53 (39.26)	202 (39.38)		
Duration of anaesthesia, Mean (SD)	159.50 (78.52)	126.33 (63.20)	168.22 (79.87)	<0.01	
Infusion and transfusion volume (mL/kg), Mean (SD)	38.49 (24.51)	28.50 (20.82)	41.12 (24.75)	<0.01	
Blood loss (mL/kg), Mean (SD)	1.75 (2.49)	1.331 (2.07)	1.85 (2.58)	0.03	
Outcomes					
Hospital stay, Median (Q1,Q3)	15 (10, 23)	12 (8.5, 20)	16 (11, 24)	<0.01	

Risk factors for different severity levels of hypothermia

Table 2 depicts a comparison of the baseline clinical data of patients with varying severity levels of hypothermia: 62.57% of cases experienced mild hypothermia (35–35.9 °C), 26.12% experienced moderate hypothermia (34–34.9 °C), and 11.31% experienced severe hypothermia (<34 °C).

Table 2 Comparison of clinical baseline data of patients with varying degrees of hypothermic severity.

Level	Overall	Mild (35–35.9 °C)	Moderate (34–34.9 °C)	Severe (<34 °C)	p	
n (%)	513	321 (62.57)	134 (26.12)	58 (11.31)		
Patient characteristics						
Sex, n (%)					0.13	
Male	304 (59.26)	201 (62.62)	71 (52.99)	32 (55.17)		
Female	209 (40.74)	120 (37.38)	63 (47.01)	26 (44.83)		
Gestational week, Mean (SD)	36.73 (3.08)	37.16 (2.89)	36.70 (2.72)	34.41 (3.86)	<0.01	
Age, mean (SD)	6.68 (6.96)	6.69 (7.26)	6.67(6.62)	6.64 (6.07)	1.00	
Weight, mean (SD)	2.72 (0.71)	2.83 (0.68)	2.70 (0.63)	2.12 (0.78)	<0.01	
Preoperative evaluation parameters					
Initial vital signs						
Preoperative temperature, Mean (SD)	36.29 (0.34)	36.30 (0.26)	36.25 (0.41)	36.29 (0.52)	0.39	
Laboratory test results						
WBC, mean (SD)	13.49 (5.96)	13.96 (6.24)	13.14 (5.44)	11.71 (5.21)	0.02	
RBC, mean (SD)	4.44 (0.82)	4.45 (0.81)	4.50 (0.80)	4.20 (0.86)	0.05	
HB, mean (SD)	153.71 (28.33)	153.86 (28.14)	156.45 (28.35)	146.60 (28.63)	0.09	
Intraoperative evaluation parameters					
Surgery type, n (%)			0.01	
General surgery	410 (79.92)	262 (81.62)	107 (79.85)	41 (70.69)		
Urology surgery neurosurgery	27 (5.26)	18 (5.61)	8 (5.97)	1 (1.72)		
Cardiothoracic surgery	54 (10.53)	24 (7.48)	16 (11.94)	14 (24.14)		
Others	22 (4.29)	17 (5.30)	3 (2.24)	2 (3.45)		
Surgical method, n (%)					0.04	
Open surgery (Thoracotomy & laparotomy)	217 (42.30)	128 (39.88)	67 (50.00)	22 (37.93)		
Endoscopic surgery (Thoracoscope & laparoscope)	201 (39.18)	122 (38.01)	51 (38.06)	28 (48.28)		
Others	95 (18.52)	71 (22.12)	16 (11.94)	8 (13.79)		
ASA, n (%)					0.15	
I/II	145 (28.27)	96 (29.91)	36 (26.87)	13 (22.41)		
III	331 (64.52)	208 (64.80)	86 (64.18)	37 (63.79)		
IV	37 (7.21)	17 (5.30)	12 (8.96)	8 (13.79)		
Planned surgery, n (%)					0.79	
No	311 (60.62)	198 (61.68)	78 (58.21)	35 (60.34)		
Yes	202 (39.38)	123 (38.32)	56 (41.79)	23 (39.66)		
Warming, n (%)					0.94	
Yes	79 (15.40)	50 (15.58)	21 (15.67)	8 (13.79)		
No	434 (84.60)	271 (84.42)	113 (84.33)	50 (86.21)		
Duration of anaesthesia, Mean (SD)	168.22 (79.87)	157.51 (78.56)	183.92 (81.84)	191.26 (72.51)	<0.01	
Infusion and transfusion volume (mL/kg), Mean (SD)	41.12 (24.75)	36.58 (22.74)	45.43 (23.11)	56.29 (31.00)	<0.01	
Blood loss (mL/kg), Mean (SD)	1.85 (2.58)	1.65 (2.27)	1.85 (2.08)	3.01 (4.38)	<0.01	
Duration of hypothermia, mean (SD)	104.05 (70.01)	84.32 (58.29)	135.93 (77.12)	139.57 (72.14)	<0.01	
Outcomes						
Hospital stay, mean (SD)	21.24 (18.05)	20.23 (18.10)	19.83 (13.31)	30.09 (24.14)	<0.01	

We observed significant differences among the three groups in multiple variables such as gestational week, preoperative weight, white blood cell (WBC) count, type of surgery, surgical method, duration of anesthesia, infusion and transfusion volume (mL/kg), blood loss (mL/kg), duration of hypothermia, and hospital stay (p < 0.05). We identified surgery type, preoperative weight, infusion and transfusion volume (mL/kg), and duration of hypothermia as independent risk factors for the severity of intraoperative hypothermia in neonates (Table 3).

Table 3 Multivariate ordered logistic regression of risk factors for the severity of hypothermia.

	OR	Lower	Higher	P	
Surgery type	1.183	1.028	1.358	0.018	
Preoperative weight	0.556	0.412	0.748	<0.001	
Duration of anaesthesia	0.997	0.993	1.000	0.073	
Infusion and transfusion volume (mL/kg)	1.011	1.001	1.022	0.018	
Duration of hypothermia	1.011	1.007	1.014	<0.001	
Note:

OR, odds ratio; CI, Confidence Interval, P < 0.05 was considered as statistically significant.

Characteristic patterns of intraoperative temperatures in neonates

We found that our patients generally attained their lowest body temperature approximately 90 min into the surgical process, with a decrease of 1–1.5 °C from their initial temperature. There was then a period of temperature stabilization characterized by a brief plateau phase, which then transitioned into a steady recovery phase (Fig. 2 depicts the temperature changes and recovery patterns of hypothermic neonates during surgery).

Figure 2 The characteristic intraoperative temperature patterns of the neonates.

Discussion

The findings of our study revealed a high incidence rate of intraoperative hypothermia in neonates (recorded at 79.17%), consistent with previous findings. Prior studies on pediatric patients have consistently documented elevated rates of intraoperative hypothermia (Schur et al., 2018). In a retrospective study on premature infants undergoing laparotomies for necrotizing enterocolitis, Sim et al. (2012) observed perioperative hypothermia in 85%. Similarly, in a retrospective study by Cui et al. (2020) the hypothermia rate in neonates was 82%. In another study, the authors reported an incidence of neonatal hypothermia as high as 83.3% (Lai et al., 2019). Similarly high incidences have also been reported in studies from developed countries (Schur et al., 2018; Kleimeyer et al., 2018). These findings emphasized the crucial need to identify risk factors for intraoperative hypothermia to apply the most effective thermal management measures and thereby enhance therapeutic outcomes for patients.

In the current study, abdominal and cardiothoracic surgeries accounted for 89.66% of cases, significantly impacting the severity of neonatal hypothermia. The type of surgery emerged as an independent risk factor for hypothermic severity. Researchers previously found that adults experienced a core temperature drop of 1.6 °C after 1 h of anesthesia, with redistribution accounting for 81% of the diminution (Sessler, 2016). The nature of abdominal and cardiothoracic surgeries that expose the body cavity to the environment further exacerbates core temperature loss. However, it is important to note that when we categorized neonates based on the degrees of hypothermia, some groups were represented by very small sample sizes. For example, only 1.72% of newborns in the urology department manifested severe hypothermia. This limitation raises the possibility that the findings might be influenced by errors attributable to the small sample sizes.

Another critical factor that we identified was preoperative weight. Neonates with lower body weights, particularly premature infants, showed a heightened susceptibility to hypothermia during surgery, and this indicated a clear link between lower weight in neonates and a higher degree of hypothermia during surgery. Newborns, especially those at lower body weight, are more susceptible to environmental temperature during surgery due to their relatively large body surface area, thin subcutaneous fat layer, large heat dissipation area, and poor insulating capability—leading to a rapid drop in body temperature (Abiramalatha et al., 2021). A smaller-scale study conducted in northern Ethiopia revealed that preterm neonates exhibited a 3.6-fold increased risk of developing hypothermia compared with full-term neonates (Shibesh et al., 2022). Although premature neonates exhibit a degree of thermoregulation (Sessler, 2016), these authors observed that neonatal body temperatures require longer to recover after reaching a plateau of hypothermia. Implementing insulating measures during surgery is therefore crucial, especially for lighter-weight preterm neonates, to reduce the risk of intraoperative hypothermia.

Consistent with prior research (Zhao et al., 2023), this study corroborated the significant influence of infusion and transfusion volumes on the incidence and severity of intraoperative hypothermia in neonates. Larger volumes of unheated infusions can substantially decrease body temperature, necessitating insulated infusion practices to mitigate the risk. Past studies indicated that the utilization of warmed intravenous fluids during surgery tended to maintain a warmer body temperature for patients relative to room-temperature fluids; and if adopted as effective, this approach can mitigate the occurrence of intraoperative hypothermia (Lau et al., 2018). In our study, although warmed irrigation fluids were routinely used, most infusions remained unheated, leading to potential hypothermia. The importance of insulated infusion devices during neonatal surgery has therefore emerged as a paramount consideration in preventing intraoperative hypothermia. We posit that this practice is considered essential for enhancing the thermal stability of neonates during surgical procedures.

We herein also discerned that the duration of hypothermia was an independent risk factor for the severity of intraoperative hypothermia in neonates. Heat generation in neonates predominantly occurs through non-shivering thermogenesis and is facilitated by neuroendocrine pathways. This mechanism is triggered by elevated activity of the sympathetic nervous system and the secretion of hormones such as thyroid-stimulating hormone, triiodothyronine, thyroxine, and norepinephrine in brown adipose tissue (Knobel & Holditch-Davis, 2010; Sessler, 2016). These hormonal and neurotransmitter activities promote the synthesis of thermogenic proteins within brown adipose tissue, notably intensifying around the 32nd week of gestation (Zhao et al., 2023). In addition, preterm infants exhibit inadequate vasomotor control at birth, lacking the capacity to activate the peripheral vasoconstriction that is crucial for heat preservation (Sessler, 2016). These investigators indicated that infants who were small or sick possessed a limited range in which their body temperature remained stable (i.e., the thermoneutral range), making them more susceptible to temperature fluctuations (Sim et al., 2012). Consequently, prolonged intraoperative hypothermia in neonates leads to substantial heat loss, exacerbating the severity of the condition. This emphasizes the critical need for vigilant monitoring and proactive thermal management strategies during neonatal surgery, to mitigate the risks associated with extended periods of hypothermia.

The detrimental effects of hypothermia are extensive, including thermal discomfort, alterations in drug metabolism and effectiveness (particularly for muscle relaxants and opioids) (Frymoyer et al., 2017), compromised platelet function and coagulatory disorders leading to increased blood loss (Rajagopalan et al., 2008; Trckova & Stourac, 2018), cardiopulmonary complications, an elevated risk of pressure ulcer development (Ahtiala, Laitio & Soppi, 2018), delayed wound healing, and an elevated likelihood of surgical site infections (Brindle et al., 2020). Exposure to cold in neonates and infants can trigger various adverse physiological responses, including increased catecholamine levels, vasoconstriction, heightened metabolism, and reduced synthesis of lung surfactant. These reactions can lead to severe health issues such as pulmonary hypertension, tissue hypoxia, arterial hypotension, metabolic acidosis from hypoperfusion, and hypoglycemia (Thomas et al., 2022). Despite perioperative warming as a standard practice recommended in evidence-based guidelines and expert consensus opinions (Forbes et al., 2009), its implementation varies significantly, particularly in many developing countries (including China); this disparity is often attributed to differences in healthcare resource allocation. Comprehensive adherence to evidence-based guidelines usually necessitates an extensive warming protocol, such as pre-warming patients to 37 °C or higher for 30 min, and using intraoperative forced-air warming covers and fluid warmers. These practices, while clinically beneficial, add to the operational costs (Forbes et al., 2009). Although routinely warming flushing fluid was a standard nursing measure used in the present study, the blood transfusions and infusions administered during surgery were not heated. Only a limited number of premature infants had access to forced-air warming devices during surgery, and there is currently no established method for preoperative pre-warming of neonates. This highlights the need for further evaluation of the appropriateness and effectiveness of perioperative warming practices. Our findings underscore the importance of managing and controlling hypothermia in intraoperative patients, and this is a focus of our future research endeavors.

Limitations

There were several additional limitations to this study. As a single-center study, the findings may not be fully representative of the broader population, potentially limiting the generalizability of our results. Surgery time for the entire population of newborns ranged from 30 to 510 min, and we only extracted data from the first 180 min, which established a large variation in the dataset. Additionally, as a retrospective study, we did not record the duration and frequency of forced-air warming, resulting in potential data exclusion that may have impacted the study outcomes.

Conclusions

Intraoperative hypothermia is a prevalent issue in neonates undergoing surgery, with a high incidence rate of 79.17%. We found that the severity of hypothermia was influenced by several risk factors, including surgery type, preoperative weight, infusion and transfusion volume, and duration of hypothermia. The nadir point of intraoperative temperature was reached approximately 90 min after the start of surgery and was then followed by a brief stabilization period and a slow recovery process. Understanding these patterns is essential for implementing targeted interventions to prevent and mitigate intraoperative hypothermia and will ultimately improve neonatal surgical outcomes.

Supplemental Information

Supplemental Information 1 Raw data.

Supplemental Information 2 STROBE checklist.

We thank LetPub for its linguistic assistance during the preparation of this manuscript.

Additional Information and Declarations

Competing Interests

Author Contributions

Human Ethics

Data Availability

The authors declare that they have no competing interests.

Kun Dai conceived and designed the experiments, performed the experiments, analyzed the data, prepared figures and/or tables, authored or reviewed drafts of the article, and approved the final draft.

Yuanling Liu conceived and designed the experiments, performed the experiments, analyzed the data, prepared figures and/or tables, authored or reviewed drafts of the article, and approved the final draft.

Lijiao Qin analyzed the data, prepared figures and/or tables, authored or reviewed drafts of the article, and approved the final draft.

Jiaxuan Mai analyzed the data, authored or reviewed drafts of the article, and approved the final draft.

Jingjing Xiao analyzed the data, authored or reviewed drafts of the article, and approved the final draft.

Jing Ruan conceived and designed the experiments, performed the experiments, prepared figures and/or tables, authored or reviewed drafts of the article, and approved the final draft.

The following information was supplied relating to ethical approvals (i.e., approving body and any reference numbers):

The Guangdong Women and Children’s Hospital granted Ethical approval to carry out the study within its facilities (Ethical Application Ref: 202301230).

The following information was supplied regarding data availability:

The raw data are available in the Supplemental File.

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
