# Peer review of "Analysis of the characteristic patterns and risk factors impacting the severity of intraoperative hypothermia in neonates"

_PeerJ, doi:10.7717/peerj.18702_

## Round 0.1 · original submission · Major Revisions

Please address the comments of the reviewers in a comprehensive revision

Reviewer 1 ·

Basic reporting

The article should be revised, my suggestions are attached.

Experimental design

The article should be revised, my suggestions are attached.

Validity of the findings

Appropriate

Additional comments

The introduction part of the article should be revised, the method should be strengthened with references, the findings should be discussed mainly in the discussion, the conclusion part is not appropriate.

Annotated reviews are not available for download in order to protect the identity of reviewers who chose to remain anonymous.

·

Basic reporting

1. Language requires major editing
2.Tables are lengthy and can be improved
3.How duration of hypothermia becomes a risk factor for hypothermia ?
4.Severity grading of hypothermia is not as per standard definitions available. (Mild between 34-36. moderate 32-24 and severe <32)

Experimental design

This is a retrospective study. Methods used appears to be Ok

Validity of the findings

As per the data, ,79.17% infants had hypothermia. This appears to be high compared to earlier published data cited by the authors. Did the authors take any corrective steps to prevent hypothermia?. If yes, those steps can be highlighted. Since the severity of hypothermia classification used by authors is not as per the standard definitions, either they can re-analyse or just give the ranges of temperature.

Additional comments

Manuscript requires major revision especially with regard to language. Some of the references listed are not as per the guidelines providded by the journal..

Reviewer 3 ·

Basic reporting

This is a great contribution to an important topic. I applaud the authors for their hard work and attempt to contribute to what is known in the field of pediatrics and neonatology surrounding this issue. While I find the discussion and conclusion clear and easy to follow, the Abstract, introduction and results could be clearer and more pointed for clarity.
The method section under abstract should be rephased for clarity and understanding. An example of a good way to introduce this study would be
A retrospective single center study was conducted and 648 subjects who had surgery performed under general anesthesia were enrolled and demographic, perioperative and intraoperative data gathered.

Experimental design

The research question is not well defined although the data collected and presented is quite compelling.
The abstract suggests and rightfully so that the study is a retrospective study designed to evaluate the characteristic patterns of intraop temperature fluctuations and risk factors for severe hypothermia. However, line 101 to 102 suggest this study was performed on neonates who received operations from 1st January 2021 to 28th February 2024. Line 118 states data gathering process was tested in October 2023.
By definition a retrospective study is done after the events have occurred. The authors received ethics committee approval to in July 2023 to collect data on neonates who have had surgery up to the dates of their approval. Authors need to clarify what exactly study duration is and if study was truly retrospective or had a prospective phase.

I suspect these are errors and can be clarified easily. However, if there was a prospective aspect or period of the study, it needs to be included and described fully. A prospective aspect will influence intraoperative nursing and anesthesiology practices that will impact results.

Validity of the findings

The authors did not include a paragraph or two on the inherent limitations of the study including: single center study, the retrospective nature of the study and approximately 50% enrollment rate (small number of subjects enrolled) from large number eligible. Although the inclusion and exclusion criteria are well laid out, 50% exclusion rate is high and can skew the results.

The authors failed to describe the location of the operating room in relation to the newborn nursery, intensive care or pediatric floor where the patients are housed prior to surgery. The paper also does not describe transport to the operating room or where surgeries are performed. All these factors could affect temperature regulation and cold exposure which will impact reproducibility of this study in other centers.

Additional comments

The results section can be better laid out to show case the major findings indicating that gestational age, chronologic age and weight impact severity of hypothermia

---

## Round 0.2 · Minor Revisions

As you can see from the reviews, you need to perform an additional edit for readability; making sure sections are in the right place in the manuscript; editing for incomplete or hanging sentences and so on

Please address these issues.

Reviewer 1 ·

Basic reporting

Clear and unambiguous, professional English used throughout.

Experimental design

Original primary research within Aims and Scope of the journal.

Validity of the findings

Impact and novelty not assessed. Meaningful replication encouraged where rationale & benefit to literature is clearly stated.

·

Basic reporting

Authors have modified the presentation. However, it requires further improvement.

Experimental design

Ok. This is a retrospective study

Validity of the findings

Ok

Additional comments

Authors need to revise the results section. Statistical tests used to be given under methods and not under results. There are multiple subheadings under results. One should only givr important findings under results.

Reviewer 3 ·

Basic reporting

Much improved from original state but still need structure and grammatical editing to improve flow, understanding and ease of reading

Experimental design

no comments

Validity of the findings

improved from original but will benefit from grammatical review

Additional comments

This version is much improved from the original but will benefit from further structural review to enhance the flow and ease of reading.

---

## Round 0.3 · accepted · Accept

Congratulations for your acceptance.

·

Basic reporting

Authors have carried out changes and improved presentation

Experimental design

Ok

Validity of the findings

Ok

Additional comments

Nil specific